# Functional Bakery Snacks for the Post-COVID-19 Market, Fortified with Omega-3 Fatty Acids

**Haralabos C. Karantonis** [1,*] **, Constantina Nasopoulou** [1] **and Dimitris Skalkos** [2]

1   Laboratory of Food Chemistry, Biochemistry and Technology, Department of Food Science and Nutrition, School of Environment, University of Aegean, Metropolitan Ioakeim 2, 81400 Mytilene, Greece; knasopoulou@aegean.gr
2   Laboratory of Food Chemistry, Department of Chemistry, University of Ioannina, 45110 Ioannina, Greece; dskalkos@uoi.gr
*   Correspondence: chkarantonis@aegean.gr; Tel.: +30-225-408-3111

**Abstract:** Flaxseed is a natural ingredient with health benefits because of its rich contents of omega-3 fatty acids and fiber. In this study, whole-meal sliced bread, chocolate cookies, and breadsticks, which were enriched with flaxseed (*Linum usitatissimu*) were produced as a natural enrichment source in order to provide functional baked goods. The three innovative products were tested as sources of omega-3 fatty acids in terms of α-linolenic acid according to EU 1924/2006 as well as for their in vitro antithrombotic/anti-inflammatory effect. The results showed that omega-3 fatty acids had high concentrations (>0.6 g per 100 g of product) in all products even after the heating treatment with constant stability during the time of consumption. All the enriched products exerted higher, but in different grade, in vitro antithrombotic/anti-inflammatory activity compared to the conventional products. The products were evaluated positively by a panel of potential consumers without significant differences compared to conventional corresponding products. Enriched bakery products with omega-3 fatty acids may represent a novel opportunity for the development of functional foods that can be locally consumed, thereby contributing to public health prevention measures that the post-COVID-19 era demands.

**Keywords:** omega-3 fatty acids; bakery snacks; sensory evaluation; in vitro nutritional functionality; antithrombotic; anti-inflammatory

## 1. Introduction

In the new post-COVID-19 period, the consumer is searching for quality, healthy foods with natural ingredients in order to protect themself from diseases, protect the environment, and provide sustainability to local economies [1–3].

Bakery snacks are an important part of the human diet and will continue to be one of the main food choices in the new era. Bread is said to be the world's earliest functional food. Functional foods are modified, enhanced, or improved foods that supply important nutrients to the human body when consumed as part of a diverse and balanced diet [4,5].

The addition of probiotics and omega-3 fatty acids to bakery products has become increasingly popular in recent years [6]. The global production of omega-3 products is estimated to be 3.3 million metric tons and worth $9.1 billion in 2018. Plant omega-3 production values are expected to grow twice as fast as marine production values during the next few years—accounting for 52% of the production value when compared to 48% for marine. In parallel, over the last few years, there have been 255 bread launches containing omega-3, which represent 7.3% of total bread launches [7,8]. Most of these have come from North America (51%), followed by Europe (21%), Asia Pacific (14%), and Latin America (14%).

At the same time linseed is a raw material that is rich in omega-3 fatty acids and fibers, making it an excellent ingredient for fortifying bakery products [9]. A legal list of



authorized nutritional claims [10] on food categorization as "source" or "high content" in omega-3 fatty acids has been established and a health claim related to essential fatty acids has been approved and placed in Commission rules [11], stating that "essential fatty acids are needed for the proper development of children".

The majority of research focuses on the experimental production of foodstuffs fortified with omega-3 fatty acids, the study of their sensory quality [12–16], and the bioavailability levels of omega-3 fatty acids in fortified foods [17–19]. Moreover, marine- or plant-originated omega-3 polyunsaturated fatty acids have been proposed as health promoting constituents for cardiovascular health.

The mechanisms though, through which those fatty acids exert their beneficial activities, are not clear [20–23]. Nevertheless, platelet activating factor (PAF) [24] has been recognized as one of the most potent lipid inflammatory and thrombotic mediators that activates various cells through its specific receptor, such as platelets [25]. Activated platelets are important contributors to thrombosis and inflammation and represent an important linkage between inflammation, thrombosis, and atherogenesis [25,26]. The in vitro inhibition of PAF-induced platelet activation from food components have been used as a research tool to investigate the nutritional functionality of those foods and their possible preventive effect against chronic disease development when consumed as part of a balanced diet [27]. Interestingly, fish oil-derived omega-3 fatty acids have been shown to suppress in vitro a fundamental process in many acute and chronic inflammatory diseases—monocyte-endothelium interaction—by inhibiting PAF activity and production [28].

The objective of the present study was to manufacture functional bakery snacks that are enriched with omega-3 fatty acids using linseed as a natural source and investigate their quality in terms of sensory evaluation, omega-3 fatty acids content, and the antithrombotic/anti-inflammatory activity through a nutritive index, as well as forecast the five-year gross sales of the products produced by local companies.

## 2. Materials and Methods

### 2.1. Manufacture of Bakery Products

#### 2.1.1. Design of the Products

The first step of the product design was to convene the HACCP food safety team of local enterprises. In this meeting, there was a full description of the products, including the necessary ingredients and raw materials, the manufacturing process of the product, the packaging and distribution, and the final characteristics of the food, as well as the documentation procedure for how to document the allergens in the nutritional details [29]. According to the legislation [30], products are not suitable for consumption for certain sensitive groups, i.e., allergic persons, therefore allergic ingredients are to be referred to on the label of the product.

#### 2.1.2. Design of Flowchart, HACCP Plan—Recipe and Manufacture of the Products

In the design of the flowchart and the confirmation of the HACCP plant in practice, three CCP (Critical Control Points) were confirmed—one for each product in the same stage of thermal processing. The critical limits were: 83 °C/1 min for sliced bread, 78 °C/40 s for cookies, and 80 °C/30 s for breadsticks [29]. During the tests of the recipes, it was decided that 50% of the linseed added to the product should be milled to reduce the feeling of the whole seed and, furthermore, to increase the stability of the fatty acids in the whole grain.

#### 2.1.3. Product Production

The creation of the three functional bakery snacks is depicted in a flow chart in Figure 1, while the ingredients for the recipes are listed in Table 1.

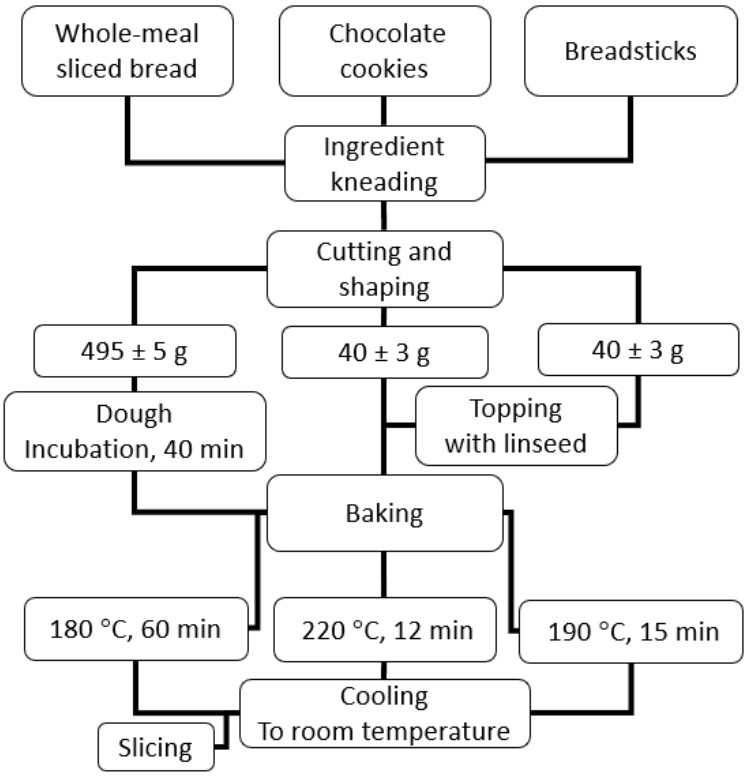

**Figure 1.** Flow chart presenting the production of the three functional bakery products.

**Table 1.** Ingredient recipes for whole-meal sliced bread, chocolate cookies, and breadsticks.

| Constituents | Whole-Meal Sliced Bread [1] | Chocolate Cookies [1] | Breadsticks [1] |
|---|---|---|---|
| Hard wheat flour (g) | 20.00 | | |
| Gluten (g) | 2.50 | | |
| Yeast (g) | 0.85 | | 1.00 |
| White wheat flour (g) | 35.00 | 45.00 | 40.00 |
| Flaxseed (g) | 16.00 | 8.00 | 12.00 |
| Sugar (g) | 0.03 | 9.00 | 5.00 |
| Salt (g) | 0.05 | 0.01 | 1.00 |
| Margarine (g) | 3.00 | 9.00 | 7.00 |
| Water (g) | 22.60 | 1.00 | 34.00 |
| Cacao (g) | | 5.00 | |
| Chocolate drops (g) | | 5.00 | |
| Egg (g) | | 10.00 | |
| Milk (g) | | 5.00 | |
| Honey (g) | | 3.00 | |

[1]: Functional bakery products enriched with flaxseed as source of omega-3 fatty acids.

### 2.2. Sensory Evaluation

Omega-3 fatty acid-enriched bakery products, namely whole-meal sliced bread, chocolate cookies, and breadsticks, were evaluated by 16 trained panelists. Each sample was evaluated on six main attributes: appearance, odour, texture, taste, aftertaste, and overall acceptance. Rating of the sensory attributes was carried out using a 9-point hedonic scale where 1 = nonexistent and 9 = too intense [11,12]. The final sensory profile of the products was determined by the average and the acceptable point (5 at the above scale) in duplicate experiments.

### 2.3. Chemical Analysis

The protein content was determined using the AOAC Kjeldahl method [31], the moisture was determined by an air oven [31], the fiber (crude) was determined by digesting the sample in a 1.25% (*v/w*) $H_2SO_4$ followed by 1.25% (*v/w*) NaOH solution [32]. Furthermore, the total fat was determined after acid hydrolysis and diethyl ether/petroleum ether extraction and fatty acids determination was performed with gas chromatography after methylation in methanol with boron trifluoride (BF3) as a catalyst [31]. The fatty acid methyl ester profiles were measured by gas–liquid chromatography on a Shimadzu 2010 chromatograph (Shimadzu Corporation, Tokyo, Japan), which was fitted with an automatic sampler AOC-20 and flame ionization detector. A fused-silica capillary column was used for the FAME analysis; DB-23, 60 m × 0.251 mm i.d., 0.25 µm (J&W, Agilent Technologies, Palo Alto, CA, USA). The oven temperature value sequence was initially 120 °C for 5 min, was raised to 180 °C at a rate of 10 °C per min, then to 220 °C at a rate of 20 °C per min, and, finally, was isothermal at 220 °C for 30 min. The injector and detector temperatures were maintained at 220 and 225 °C, respectively. The carrier gas was a high purity helium with a linear flow rate of 1 mL per min and a split ratio of 1:50. The individual FAMEs were identified by comparison with the relative retention time of the FAMEs peaks from the samples with the standard mixtures 37 Component FAME Mix (47885-U Supelco, Bellefonte, PA, USA) and Qualmix Fish S (89-5550 Larodan Fine Chemicals AB, Malmö, Sweden). The carbohydrates and energy were calculated from proximate analysis values.

All products were subjected to the determination of omega-3 fatty acids before and after baking in order to assure their availability after baking.

### 2.4. Shelf-Life Determination

The chemical and sensorial shelf life of the enriched bakery products was determined by means of sensory scoring and omega-3 content to meet the legislative limits for nutritional claims for products kept at room temperature for 24 days. The measurements were performed on days 0, 4, 7, 18, and 24. The overall market shelf life of the products was determined as the combination of the chemical and sensory values.

### 2.5. Anti-Thrombotic and Anti-Inflammatory Activity

All chemical reagents and solvents were of analytical grade and were supplied by Merck (Darmstadt, Germany). Platelet-activating factor (β-Acetyl-γ-*O*-hexadecyl-*L*-α-phosphatidylcholine hydrate) and bovine serum albumin (BSA) were obtained from Sigma (St. Louis, MO, USA).

### 2.5.1. Lipid Extraction

An amount of 15.0 mL of methanol was mixed with 3.0 g of flour sample and the mixture was agitated at 200 rpm on a GFL 3017 orbital shaker (GFL, Burgwedel, Germany) for 15 min. After that, 7.5 mL of chloroform was added to the mixture, which was followed by agitation at 200 rpm for 15 min. In the next step, after taking into consideration the moisture content of the samples, 6.0 mL of either distilled water (first version of lipid extraction) or 1 m aqueous sodium chloride solution 0.5% in acetic acid (second version of lipid extraction) was placed in the mixture in order to achieve a ratio for the solvents of methanol/chloroform/water that was equal to 1/2/0.8 (*v/v/v*), and then an extra agitation was performed at 200 rpm for 15 min. Then, depending on the version of the method, 7.5 mL of methanol were added along with either 7.5 mL of distilled water (first version) or 7.5 mL of 1 m aqueous sodium chloride solution 0.5% in acetic acid (second version), thereby achieving a ratio for the solvents of methanol/chloroform/water that was equal to 1/1/0.9 (*v/v/v*). After a final agitation at 200 rpm for 15 min, the samples were stored overnight at 4 °C. The samples were then centrifuged for 5 min at 2000× *g* in a Hermle Z 383 centrifuge (Hermle Labortechnik, Wehingen, Germany) and the lower phase of the biphasic solvent system was collected into a pre-weighed glass flask along with a 5.0 mL chloroform rinse of the upper phase. The samples were dried using a Lab Tech

EV 311 Rotary evaporator (Lab Tech, Milan, Italy), which was weighed on a KERN ABJ analytical balance (Kern and Sohn GmbH, Balingen, Germany), suspended in 2.0 mL of chloroform/methanol: 1/1 (*v/v*), and stored at −40 °C until further study.

### 2.5.2. In Vitro Anti-Thrombotic and Anti-Inflammatory Activity

The in vitro antithrombotic and anti-inflammatory activity of the lipid extracts that were enriched in omega-3 fatty acids or conventional food products were evaluated on a Chrono-Log 500-Ca aggregometer (Chrono-Log Co., Havertown, PA, USA) that was connected to a computer (Aggro/Link software; Chrono-Log, Hawertown, PA, USA) according to their ability to inhibit the thrombotic and inflammatory lipid mediator of PAF towards platelet rich plasma (PRP) [33]. Aliquots of dissolved lipid extracts that were enriched with omega-3 fatty acids or conventional food products and PAF solution were evaporated under a stream of nitrogen and reconstituted in BSA (2.5 mg/mL saline). The platelet response induced by PAF ($10^{-7}$ M, final concentration) was measured in PRP before (considered as 0% inhibition) and after the addition of various concentrations of the examined sample. Consequently, the plot of the percentage inhibition (ranging from 20 to 80%) versus the different concentrations of the sample is linear. From this curve, the amount of lipids required for a 50% inhibition (inhibitory amount for 50% inhibition; IA50) against PAF was calculated and expressed in μg for the lipid extracts.

### 2.6. Marketing Plan: Development

The 5-year marketing plan in this research was investigated in order to target the objectives for the gross sales of the products under study by a local bakery and cheese company that were involved. The plan was based on sales: (a) in the first two years they would be at the local markets, where the company is active and well-known; (b) in the third- and fourth-year there would be expansion to the national market; and (c) in the fifth year there would be promotion to the European market, targeting two or three specific countries such as Germany, France, and England. Data analysis was based on the last three years of the company's performance gross sales for its conventional products that were successfully promoted in the local and national Greek market. In addition, the overall company's performance and its plans for growth and expansion in new areas of activities—such as healthy food snacks—were considered when formulating the specific marketing plan. Market trends were also considered at the global level by evaluating existing marketing data on omega-3 functional foods in general and on omega-3 bakery products specifically. The formulated plan considered conventional activities such as advertising, exhibitions, events, etc., as well as modern means such as social media, internet usage, etc. for the promotion of the products and their health claims. Based on the proposed activities each year, and combined with the Greek and global market trends, the feasible six-year projected sales were forecasted and presented in the section below.

### 2.7. Statistics

A comparison of the results was performed using the statistical package SPSS *v.*21 using the average value comparison assay for independent samples (Student's t-test for independent samples) at a significance level equal to 0.05.

## 3. Results

### 3.1. Manufacture of Bakery Products

A representative photo of the three products is given in Figure 2.

### 3.2. Sensory Evaluation of the New Products

The organoleptic properties and acceptance of the new products are presented in Figure 3. The overall acceptability is above the set limit for acceptance (5.00) and high enough (7.29 and 7.86 for whole-meal sliced bread and chocolate cookies, followed by 6.36

for breadsticks) to indicate that their fate in the market could be successful from a sensorial point of view.

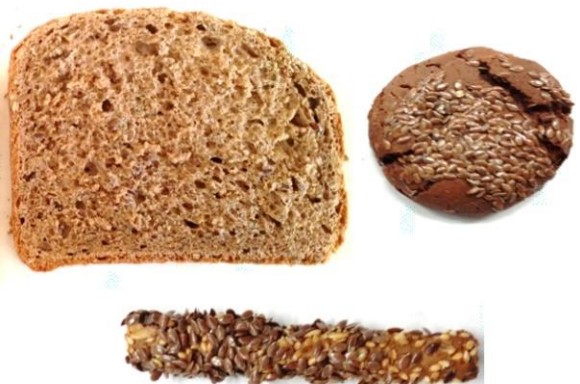

**Figure 2.** Representative photos for the three functional bakery snacks.

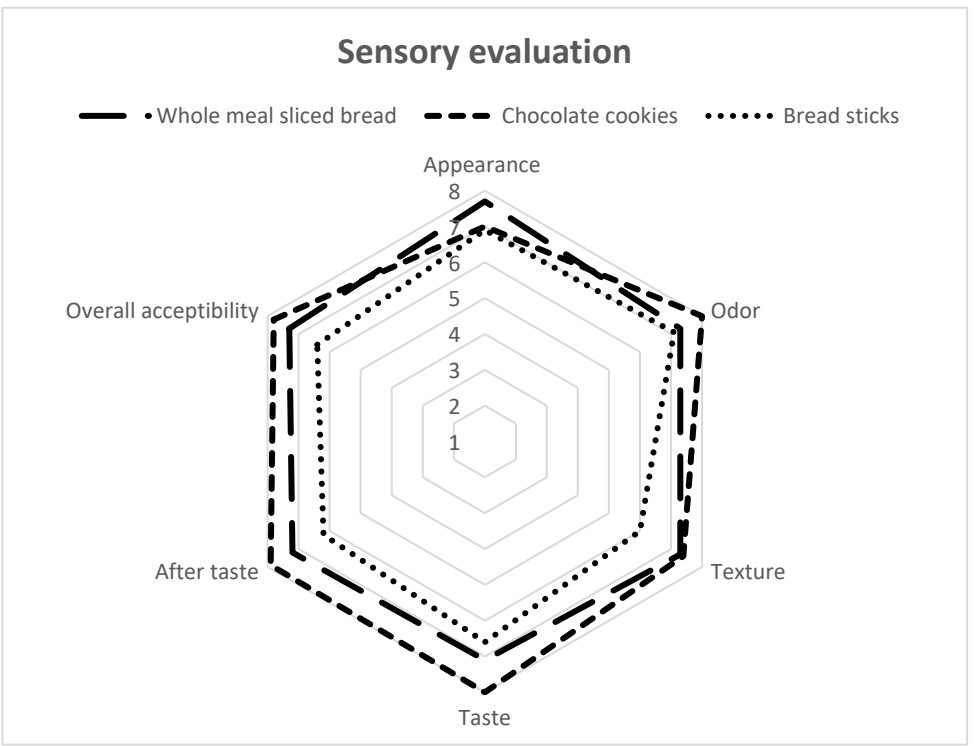

**Figure 3.** Sensory evaluation of whole-meal sliced bread, chocolate cookies, and breadsticks. Results are expressed as a mean of the responses from 16 trained panelists.

### 3.3. Chemical Analysis of the Products

The nutritional composition of the new products is depicted in Table 2. According to Regulation 1924/2006 on nutritional claims [10], whole-meal sliced bread is a "source of fiber" since it contains 5.4 g fibers/100 g of product, while the limit for this claim is 3 g fibers/100 g of product and, furthermore, it is a "source of protein" since $16.7 \pm 100\%$ of the energy of the product comes from proteins, while the respective limit for this claim is 12% according to the formula: % energy from protein s = [(g proteins/100 g × 4 kcal/g)/(total energy/100 g)] × 100. Moreover, chocolate cookies ($7.03 \pm 0.30$ g/100 g product) and breadsticks ($7.08 \pm 0.12$ g/100 g product) are both a "source of fiber".

**Table 2.** Nutritional composition of whole-meal sliced bread, chocolate cookies, and breadsticks.

| Constituents | Whole-Meal Sliced Bread [1] | Chocolate Cookies [1] | Breadsticks [1] |
|---|---|---|---|
| Fat (g/100 g) | 10.0 ± 0.83 | 17.52 ± 1.39 | 15.33 ± 0.78 |
| Humidity (g/100 g) | 29.80 ± 1.12 | 10.29 ± 0.49 | 8.41 ± 0.39 |
| Proteins (g/100 g) | 12.30 ± 0.59 | 10.03 ± 0.42 | 6.77 ± 0.33 |
| Carbohydrates (g/100 g) | 39.40 ± 1.69 | 50.90 ± 0.15 | 57.98 ± 1.38 |
| Ash (g/100 g) | 3.80 ± 0.13 | 4.20 ± 0.10 | 4.62 ± 0.11 |
| Fibers (g/100 g) | 5.40 ± 0.21 | 7.03 ± 0.30 | 7.08 ± 0.12 |
| Energy Kcal/100 g | 294.74 ± 2.36 | 401.50 ± 2.09 | 396.97 ± 2.63 |

[1]: Each result is the mean value of triplicate experiment.

The results of the effect of baking on the content of omega-3 fatty acids are shown in Table 3. According to the results, baking does not influence the omega-3 fatty acids content of the final products.

**Table 3.** Effect of baking on omega-3 fatty acids concentration.

| Constituents | Whole-meal Sliced Bread [1] | | Chocolate Cookies [1] | | Breadsticks [1] | |
|---|---|---|---|---|---|---|
| | Before | After | Before | After | Before | After |
| Total fat (g/100 g) | 10.90 ± 0.99 | 10.01 ± 0.83 | 18.12 ± 1.34 | 17.74 ± 1.39 | 15.99 ± 0.96 | 15.33 ± 0.78 |
| Fatty acids (g/100 g fat) | | | | | | |
| Myristic acid C14:0 | 0.50 ± 0.09 | 0.60 ± 0.11 | 1.00 ± 0.08 | 0.90 ± 0.07 | 0.20 ± 0.01 | 0.20 ± 0.01 |
| Palmitic acid C16:0 | 18.90 ± 2.91 | 22.70 ± 2.99 | 25.30 ± 0.95 | 25.00 ± 1.11 | 8.30 ± 0.40 | 7.90 ± 0.03 |
| Stearic acid C18:0 | 7.80 ± 0.30 | 9.20 ± 0.41 | 6.90 ± 0.31 | 6.80 ± 0.31 | 2.90 ± 0.19 | 2.90 ± 0.15 |
| Oleic acid C18:1n9c | 24.30 ± 1.51 | 25.40 ± 1.67 | 33.00 ± 2.11 | 31.10 ± 2.43 | 25.30 ± 1.21 | 24.40 ± 1.69 |
| Vaccenic acid C18:1n7c | 0.70 ± 0.03 | 0.70 ± 0.04 | 0.90 ± 0.051 | 0.90 ± 0.05 | 0.80 ± 0.07 | 0.76 ± 0.06 |
| Linoleic acid C18:2n6c | 24.60 ± 1.98 | 23.10 ± 0192 | 22.50 ± 1.01 | 22.70 ± 0.99 | 41.70 ± 2.56 | 38.10 ± 2.68 |
| α-Linolenic acid C18:3n3c | 20.40 ± 1.49 | 19.60 ± 1.79 | 9.10 ± 0.72 | 9.50 ± 0.61 | 18.30 ± 1.08 | 19.00 ± 1.23 |

[1]: Results from experiments, which were run in triplicate, are expressed as mean value ± standard deviation.

Based on the results and Regulation 1924/2006 on nutritional claims [10], whole-meal sliced bread has a "high content" of omega-3 fatty acids since their concentration is 1.96 ± 0.18 g/100 g of the product, while the respective limit for this nutritional claim is 0.6 g/100 g of the product. Moreover, chocolate cookies and breadsticks also have a "high content" of omega-3 fatty acids (0.95 ± 0.06 and 1.90 ± 0.12 g/100 g product, respectively).

### 3.4. Shelf-Life Assessment

The shelf life, both sensorial and chemical, of the new products is presented in Table 4. The shelf life, concerning the minimum concentration of omega-3 fatty acids for the nutritional claim, is 18 days at room temperature for whole-meal sliced bread and more than 24 days for chocolate cookies and breadsticks. The sensorial shelf life is 7 days at room temperature for whole-meal sliced bread and 24 days for chocolate cookies and breadsticks. Given this, the combined shelf life of the new products at room temperature is 7 days for whole-meal sliced bread and 24 days for chocolate cookies and breadsticks, which is acceptable for the food market.

### 3.5. Lipid Extraction and In Vitro Antithrombotic and Anti-Inflammatory Activity

In order to optimize lipid extraction, two versions of the Bligh and Dyer method [34] for lipid extraction were evaluated for their capacity in enriched flour samples in triplicate experiments. The results showed that the second version that used 1 m aqueous sodium chloride solution 0.5% in acetic acid instead of distilled water for the first version extracted lipids more efficiently. More specifically, 123.3 ± 8.0 mg of lipids per g of flour were extracted by performing the second version of the extraction versus 106.7 ± 7.2 mg of lipids per g of flour, which were extracted by performing the first version of the extraction ($p < 0.05$). The procedure of the second version was further applied to the prepared

foods to extract their total lipids in order to evaluate their in vitro antithrombotic and anti-inflammatory activities.

**Table 4.** Monitoring of sensorial attributes and omega-3 fatty acid concentration.

| Attribute | Product | Day 0 | Day 4 | Day 7 | Day 18 | Day 24 |
|---|---|---|---|---|---|---|
| Appearance [1] | Whole-meal sliced bread | 7.7 ± 0.3 | 8.1 ± 0.3 | 7.8 ± 0.3 | Na [2] | NA |
| | Chocolate Cookies | 7.1 ± 0.4 | 7.8 ± 0.3 | 7.0 ± 0.3 | 6.5 ± 0.3 | 5.3 ± 0.2 |
| | Breadsticks | 6.9 ± 0.3 | 6.9 ± 0.3 | 6.8 ± 0.3 | 6.6 ± 0.2 | 5.8 ± 0.2 |
| Odor [1] | Whole-meal sliced bread | 7.3 ± 0.4 | 8.0 ± 0.3 | 7.7 ± 0.3 | NA | NA |
| | Chocolate Cookies | 8.1 ± 0.3 | 8.0 ± 0.3 | 7.5 ± 0.3 | 6.8 ± 0.2 | 6.0 ± 0.3 |
| | Breadsticks | 7.1 ± 0.4 | 7.3 ± 0.4 | 6.8 ± 0.3 | 6.1 ± 0.3 | 5.4 ± 0.3 |
| Texture [1] | Whole-meal sliced bread | 7.3 ± 0.3 | 7.3 ± 0.4 | 7.4 ± 0.4 | NA | NA |
| | Chocolate Cookies | 7.4 ± 0.3 | 7.9 ± 0.4 | 6.5 ± 0.3 | 6.0 ± 0.2 | 5.1 ± 0.2 |
| | Breadsticks | 6.0 ± 0.3 | 7.4 ± 0.4 | 7.3 ± 0.4 | 6.6 ± 0.3 | 5.3 ± 0.2 |
| Taste [1] | Whole-meal sliced bread | 7.1 ± 0.3 | 6.9 ± 0.3 | 7.1 ± 0.4 | NA | NA |
| | Chocolate Cookies | 8.0 ± 0.4 | 7.9 ± 0.4 | 7.5 ± 0.4 | 6.8 ± 0.3 | 5.4 ± 0.3 |
| | Breadsticks | 6.6 ± 0.3 | 7.6 ± 0.3 | 7.2 ± 0.3 | 5.9 ± 0.2 | 5.2 ± 0.2 |
| After taste [1] | Whole-meal sliced bread | 7.2 ± 0.2 | 7.2 ± 0.2 | 7.3 ± 0.3 | NA | NA |
| | Chocolate Cookies | 7.9 ± 0.2 | 7.8 ± 0.2 | 7.2 ± 0.3 | 6.6 ± 0.3 | 5.2 ± 0.1 |
| | Breadsticks | 6.2 ± 0.3 | 7.7 ± 0.2 | 7.0 ± 0.3 | 5.8 ± 0.3 | 5.0 ± 0.3 |
| Overall acceptability [1] | Whole-meal sliced bread | 7.3 ± 0.4 | 7.3 ± 0.3 | 7.3 ± 0.3 | NA | NA |
| | Chocolate Cookies | 7.9 ± 0.3 | 7.9 ± 0.3 | 7.2 ± 0.3 | 6.5 ± 0.2 | 5.2 ± 0.3 |
| | Breadsticks | 6.4 ± 0.3 | 7.3 ± 0.3 | 7.1 ± 0.4 | 5.9 ± 0.3 | 5.1 ± 0.3 |
| α-Linolenic acid C18:3n3c (g/100 g product) [3] | Whole-meal sliced bread | 1.3 ± 0.2 | 1.0 ± 0.1 | 1.0 ± 0.1 | 0.59 ± 0.1 | NA |
| | Chocolate Cookies | 1.3 ± 0.2 | 0.9 ± 0.1 | 1.1 ± 0.1 | 0.9 ± 0.1 | 1.2 ± 0.1 |
| | Breadsticks | 2.9 ± 0.1 | 6.9 ± 0.2 | 7.2 ± 0.3 | 6.8 ± 0.4 | 6.2 ± 0.3 |

[1] Results are expressed as mean ± standard deviation of the responses from 16 trained panelists; [2] NA: not analysed because mold has been grown on the product; [3] Results are expressed as mean ± standard deviation from triplicate experiment.

The results from the in vitro inhibition of PAF-induced platelet activation are presented in Table 5. Flour enriched in omega-3 fatty acids, whole-meal sliced bread, chocolate cookies, and breadsticks showed higher antithrombotic and anti-inflammatory activities compared to the corresponding conventional samples. More specifically, the enriched flour, sliced bread, chocolate cookies, and breadsticks showed ×4.7, ×4.6, ×1.1, and ×1.6 higher activities compared to their respective conventional samples. The results show that the enrichment of omega 3 fatty acids may result in food products with increased nutritional value, but this enrichment may be affected by the processing that is followed to prepare each product.

**Table 5.** In vitro antithrombotic and anti-inflammatory activity.

| Food Product | [1] IA$_{50}$ | |
|---|---|---|
| | Conventional | Enriched |
| Flour | 7.60 ± 0.08 | 1.61 ± 0.07 [2] |
| Whole-meal sliced bread | 86.13 ± 1.01 | 18.73 ± 0.40 [2] |
| Chocolate cookies | 55.72 ± 0.63 | 49.72 ± 0.60 [2] |
| Breadsticks | 73.92 ± 0.83 | 46.72 ± 0.73 [2] |

[1]: Inhibitory amount for 50% inhibition of PAF activity toward washed platelets expressed in μg of lipid extracts. Values from experiments in triplicate are expressed as mean ± standard deviation. [2]: Statistically significant difference of enriched versus conventional products at significance level equal to 0.05.

Lipid mediators are a heterogeneous group of molecules that mediate several physiological cellular functions that, when they are dysfunctional, may lead to the development of many chronic diseases. Food-derived bioactive compounds can beneficially influence metabolism, thereby offering an attractive way to prevent the establishment of chronic diseases.

An increased intake of omega-3 fatty acids from foodstuffs is related to a favorable clinical profile of various chronic diseases. Platelet activating factor (PAF) is one of the most potent inflammatory and thrombotic lipid mediators, playing a crucial role in the

initiation and propagation of atherosclerosis. Therefore, the omega-3 fatty acid-induced PAF inhibition is very important in terms of their nutritional value.

*3.6. Marketing Plan*

The results of the marketing plan developed in terms of forecasted productions and sales over the first 5 years are shown in Table 6. The first two years, when the products will be promoted to the local market, project that the sales will have a 102.7% annual increase, reaching 75,000€. A 120% sales increase is expected in the third year due to the promotion of the products in the overall Greek market, which will be followed by a 45% medium increase the coming year within the same market. Finally in the last year, with the promotion of the products to international markets, an initial 20% increase in sales is expected, which will increase steadily by 20–30% annually in the years to come based on more promotion internationally. A similar annual increase is expected for the production capacity from year to year as well: 2nd from 1st year (103.3%), 3rd from 2nd year (131.7%), 4th from 3rd year (42.1%), and 5th from 4th year (20.9%). The expected differences in production and sales between the three products over the years are based on the company's production and sales for the corresponding conventional products, which are already on the market.

**Table 6.** Expected productions/sales of the new omega-3 bakery products.

|  |  | Whole-Meal Sliced Bread | Chocolate Cookies | Breadsticks | TOTAL |
|---|---|---|---|---|---|
| 1st year [1] | Quantity(Kg) | 6600 | 2500 | 3000 | 12,100 |
|  | Sales (€) | 10,000 | 12,000 | 15,000 | 37,000 |
| 2nd year [1] | Quantity(Kg) | 13,300 | 53,000 | 6000 | 24,600 |
|  | Sales (€) | 20,000 | 25,000 | 30,000 | 75,000 |
| 3rd year [2] | Quantity(Kg) | 33,300 | 11,700 | 12,000 | 57,000 |
|  | Sales (€) | 50,000 | 55,000 | 60,000 | 165,000 |
| 4th year [2] | Quantity(Kg) | 46,000 | 17,000 | 18,000 | 81,000 |
|  | Sales (€) | 70,000 | 80,000 | 90,000 | 240,000 |
| 5th year [3] | Quantity(Kg) | 56,000 | 20,000 | 22,000 | 98,000 |
|  | Sales (€) | 84,000 | 96,000 | 108,000 | 288,000 |

[1]: Sales at the local market only; [2]: Sales at the Greek market as well; [3]: Sales in targeted European countries as well.

The proposed marketing plan predicts 288,000€ expected annual sales in the 5th year from the three selected products. This amount, for a company of 5,000,000€ total sales with hundreds of products produced, is a satisfactory sales target from three products alone (5.7%), and a good driving force for the company to invest in for the implementation of the five-year proposed marketing plan.

## 4. Discussion

The relevance and innovation of the present study lies in the incorporation of omega-3 fatty acids as an ingredient for manufacturing healthy bakery snacks that offer new opportunities for the local Greek snack market. The three bakery snacks manufactured are rich in omega-3 fatty acids since their content is higher than 0.6 g of omega-3 fatty acids per 100 g of baked product, and they are also a source of dietary fiber since their content is higher than 3 g of fiber per 100 g of final product. Moreover, whole-meal sliced bread is, in addition, a source of protein since more than 12% of the % energy comes from it.

The produced whole-meal sliced bread, chocolate cookies, and breadsticks have 2.0 g, 1.7 g, and 2.9 g of -linolenic acid, respectively, and daily bread consumption of 160 g [35,36] is higher than that of the other two goods that are taken on a more irregular basis.

Taking into account the frequency and quantity of an adult's average consumption of manufactured products—given that they need 2.22 g of a-linolenic acid per day [37]—

whole-meal sliced bread looks to practically meet the daily requirements for a-linolenic acid, being a better source of omega-3 fatty acids than chocolate cookies or breadsticks.

Omega-3 fatty acids incorporation also enhanced the antithrombotic and anti-inflammatory bioactivity in terms of the in vitro inhibition of platelet activating factor against platelet rich plasma, thereby indicating health protective properties compared to the conventional formulations.

Omega-3 fatty acid-enriched bakery products were well accepted by panelists for their sensorial attributes.

The findings of this study demonstrated that omega-3 fatty acids might represent a valuable ingredient to improve the nutritional and health-protective properties of bakery snacks. Indeed, linseed (*Linum usitatissimum* L.)—generally known as flaxseed—has been demonstrated to boost immunity against viral infections as well as control cytokine storms and inflammatory mediators [38–43]. Moreover, due to the high number of both soluble and insoluble dietary fibers, flaxseed is beneficial for gut health, thereby boosting the immune system [44,45].

### 5. Conclusions

In conclusion, the design and preparation of enriched bakery snacks with flaxseed as a source of omega-3 fatty acids may represent a novel opportunity for the development of functional foods that are sustainable in the food market and could also contribute to the prevention of public health issues when consumed.

**Author Contributions:** Conceptualization, supervision, methodology H.C.K. and D.S.; investigation, writing—original draft preparation and writing—review and editing H.C.K., C.N. and D.S.; software, data curation and resources H.C.K. All authors have read and agreed to the published version of the manuscript.

**Funding:** This research received no external funding.

**Institutional Review Board Statement:** Not applicable.

**Informed Consent Statement:** Not applicable.

**Data Availability Statement:** Not applicable.

**Conflicts of Interest:** The authors declare no conflict of interest.

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
