# Peer review of "Functional Bakery Snacks for the Post-COVID-19 Market, Fortified with Omega-3 Fatty Acids"

_sustainability, doi:10.3390/su14084816_

Round 1

Reviewer 1 Report

.

Reviewer 2 Report

The objective of the present study was to manufacture healthy innovative bakery 86 snacks enriched with omega-3 fatty acids using linseed as a natural source and investigate their quality.

The manuscript "Innovative bakery snacks, enriched with omega-3 fatty acids using natural sources for the new post COVID-19 market" by Soumeia Karantonis et al.  is interesting and well presented research that might be useful for future readers. However, certain additional minor corrections should performed before publication.

Page 7. The results in Table 3 are repeated and are mentioned in Table 1, so it is sufficient to refer to Table 1 when commenting or discussing them.

Page 7. It was better to represent Table 4 for sensory evaluation by a sensory map, as it gives a clearer visual evaluation.

Page 10. Discussion

I regret that the discussion was weak and the results obtained were not discussed, and if it referred to the importance of foods rich in omega-3, what I presented remains general and does not focus on the current study. Please rewrite the discussion in a manner consistent with the results obtained. Keeping the last paragraph because it's good.

Reviewer 3 Report

Overall, it's a short and interesting job. However, the lack of product recipes strongly lowers its scientific significance, shifting the work towards more economic, perhaps a little practical, work. In general, the work presents interesting results in terms of the possibility of supplementing products with fatty acids and is of significant informational importance due to the fact that they are quite widely consumed products. However, I suggest adding recipes. It seems to me that the authors also associate their work too closely with COVID-19, such as Lines 30-31: I don't think the search for new and healthy products is directly related to covid-19. Similar references appear in the discussion and suggest that it is currently the most important factor in a healthy lifestyle, which in my opinion is a misunderstanding. Please correct these formulations, it may be one of the factors, but probably not the most important one.

The conclusions are too brief. I also suggest extending the conclusions. For example, to find out what type / products the authors consider to be the best possible source of omega-3 acids.

Round 2

Reviewer 1 Report

"3. Conclusion "should be numbered according to the order of 3.1, 3.2, 3.3. In order to improve the quality of the manuscript, we hope it can be revised.

Author Response

According to the reviewer’s comment we proceeded to improve the way in which the conclusions are reported by giving the correct numbering to the paragraphs of the result section and we also modified the numbering to match the order in which the individual topics are mentioned in the methodology section, that is:

3.1 Manufacture of bakery products 3.2 Sensory evaluation of the new products

3.3 Chemical analysis of the products

3.4 Shelf-life assessment

3.5 Lipid Extraction and in vitro anti-trhombotic and anti-inflammatory activity

3.6 Marketing plan

Reviewer 2 Report

I think the authors have made the required revisions.

Author Response

We thank the reviewer  for taking the time to assess our revised manuscript manuscript